# Exploring National Transportation Safety Board Aviation Modality Recommendations Through Content and Sentiment Analyses: 2015–2019

**Brian J. Roggow** [ID]

Robertson Safety Institute, Embry-Riddle Aeronautical University, 3700 Willow Creek Rd.,
Prescott, AZ 86301, USA; brian.roggow@erau.edu; Tel.: +1-928-777-4744

**Abstract:** Aviation safety recommendations are the National Transportation Safety Board's key mechanism for effecting improvements and curtailing subsequent accidents. Aviation safety recommendations and their associated correspondence have been minimally explored in the extant literature, potentially overlooking constrained versus successful risk mitigation themes. This research aimed to qualitatively explore 187 aviation safety recommendations using a framework adapted from the SHELL model. The research also examined the recommendations' correspondence content to illuminate the characteristics typical of positive versus negative sentiments. The results included risk mitigation themes distributed across the categories of addressees, report statuses, and reiterations. Addressing company, management, manning, or regulatory issues was the most prevalent risk mitigation strategy, followed by physical environment and other human-system support mitigations. The sentiment analyses' results included distributions across addressees, statuses, time, reiterations, and correspondences. NTSB and addressee correspondence sentiments remained mostly consistent over time and interactions, whereas differences were observed based on addressees and unacceptable report statuses. This article offers the first systematic analysis of NTSB aviation safety recommendations' risk mitigation themes and addressee correspondences.

**Keywords:** aviation safety; safety recommendations; risk mitigation; accident investigation; content analysis; sentiment analysis; government communication; stakeholder engagement

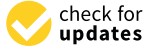

## 1. Introduction

The National Transportation Safety Board (NTSB) investigates all United States civil aircraft accidents and serves as an accredited representative to foreign investigations under the provisions of the International Civil Aviation Organization (ICAO) Annex 13; their investigation processes can span from several months to several years [1]. The NTSB issues final accident reports that include factual information, analyses, conclusions, and, when applicable, recommendations. Recommendations may also be issued from incident investigations or safety research reports. Recommendations are the NTSB's primary means of effecting safety improvements in all modes of transportation, including aviation [1,2]. Since its inception in 1967, the NTSB has issued over 15,500 safety recommendations, nearly 6000 of which have been for the aviation modality [1].

The NTSB's Case Analysis and Reporting Online (CAROL) is a database repository and search tool for investigations and safety recommendations across all transportation modes [1]. This robust tool enables custom queries about recommendations' textual content and numerous data fields, supporting public access to significant safety information. The

NTSB's recommendations include categorical statuses, including open and closed, with varying secondary descriptors (e.g., exceeds, acceptable, alternative, unacceptable, and reconsidered). Furthermore, CAROL includes each safety recommendation's correspondence between the NTSB and addressee(s) from a recommendation's issuance through closure.

From 1990 through 2023, the NTSB Most Wanted List (MWL) advocated for key improvements within and across transportation modes [3]. Sweedler [2] communicated that safety issues were considered for placement on the MWL if the issue carried significant potential for improvements at the national level, were of—or needed—high public visibility, and could be completed in a timely manner. However, the path of individual safety recommendations to the MWL's broader focus on unresolved issues was not always clear; further, several issues remained on the MWL for multiple years, raising questions about the timely completion criterion. Recommendations that are not successfully implemented may result in repeated accidents and incidents, inhibiting the national airspace system's safety.

Since the MWL's retirement, the NTSB has illuminated safety issues through a recommendations spotlight webpage. Each month, the NTSB [4] highlights a few recommendations that have been implemented successfully across each mode of transportation. Intriguingly, their approach has shifted from highlighting the MWL's unresolved issues to recent successes. Nonetheless, scientific research on MWL content, recent safety spotlights, and broader recommendation attributes is scarce. Sweedler [2], formerly an NTSB Office of Safety Recommendations executive, examined 25 years of NTSB recommendations, providing context, selected descriptive statistics, and noteworthy examples.

### 1.1. Research Purpose

Since Sweedler's [2] examination, comprehensive and scientific examinations of NTSB recommendations have been scarce. Beyond the NTSB's [1] "recommendations at a glance" descriptive statistics, the nature of the NTSB's recommendation content has been insufficiently explored. Furthermore, the nature of addressee interactions has not been systematically explored. Therefore, this research aimed to close these gaps.

This study employed qualitative methods and had two main goals. First, the study aimed to explore NTSB aviation recommendations' prevailing risk mitigation content themes. To accomplish this goal, an ICAO [5] Accident/Incident Data Reporting (ADREP) taxonomy, modeled from Hawkins' [6] SHELL model, was adapted to categorize the recommendations' content. Qualitative analyses of the NTSB recommendations' content elucidated new information about prevailing safety issues.

Second, the study aimed to explore the nature of interactions between the NTSB and recommendation addressees through sentiment analyses. Recommendation addressees include organizations best equipped to resolve safety issues, including the U.S. Department of Transportation, the Federal Aviation Administration (FAA), other federal and state agencies, manufacturers, operators, labor unions, and industry and trade organizations [1]. The objectives of the sentiment analyses included differentiations among (1) addressees, (2) communication correspondence counts, (3) the duration of time since the recommendation's issuance, and (4) recommendation statuses. NVivo qualitative data analysis software supported the exploration of these factors.

### 1.2. Delimitations and Assumptions

Inclusively, the five years spanning from 2015 to 2019 were selected due to their completeness, given that there are no open investigations within this period and, therefore, no impending recommendations. This completeness was balanced against recency and the continued relevancy of the recommendations in focus. At the onset of this study, open investigations spanned from 2022 through 2024; the inclusion of these calendar years would

have resulted in an inconsistent and incomplete dataset. While the NTSB issues safety recommendations across all transportation modes, aviation was this study's sole domain of interest.

This study assumed that the NTSB has reliably retained aviation safety recommendations and addressee correspondence. Further, NTSB final report recommendations were assumed to be wholly and consistently populated in the NTSB CAROL database; database entry or retainment errors would negatively influence the results' validity. Qualitative data analyses utilizing interactive coding relied on the researcher's consistent and complete applications of coding schemes [7]. This study assumed that the coding of textual data was consistent; however, intra-rater reliability, or consistency, was not explicitly measured.

*1.3. Review of Relevant Literature*

This section examines a sample of extant research related to NTSB safety recommendations. The section begins with a discussion of accident investigation and safety recommendation research to contextualize this study's focus in contrast to previous efforts. Next, the content concisely synthesizes studies that have qualitatively explored aircraft accident data using the NVivo software, contextualizing its use for the current study. Thereafter, a sample of research exploring NTSB communications content is presented to illuminate the gap of systematic sentiment analyses. This section concludes with the background for the coding framework employed by the current study.

1.3.1. Accident Investigation and Recommendation Research

While recommendations are the most effective means of effecting safety improvements, research has often focused on accident causation factors. Wiegmann et al.'s [8] examination of 14,436 general aviation accidents classified causal and contributing factors through the Human Factors Analysis and Classification System (HFACS). Wiegmann et al.'s [8] results provided comprehensive and interesting insights; however, the findings were not contextualized alongside the investigations' recommendations, leaving uncertainty about the novelty of their conclusions compared to the NTSB's. Nonetheless, Wiegmann et al. [8] aptly identified the critical need to methodically shape and evaluate interventional mitigations.

Sweedler [2] reported that the NTSB issued nearly 9000 safety recommendations to over 1250 addressees between 1967 and 1995, and over 80 percent were accepted and implemented. Sweedler's assessment explored notable recommendations that resulted in critical aviation safety improvements, such as ground proximity warning system (GPWS) installation in air carrier aircraft. Relatedly, Waycaster et al. [9] assessed safety improvements implemented between 2002 and 2009 across transportation modes. While commercial aviation transportation was identified among the safest modes of transportation, Waycaster et al. [9] identified that commercial aviation regulatory enhancements far exceeded other modes of transportation, partly due to investigations of significant fatal aircraft accidents and the subsequent issuance and implementation of recommendations. These findings illuminate the critical importance of effectively shaping safety policies and support this study's exploration of NTSB safety recommendations.

Nonetheless, a modest percentage of recommendations have been closed with unacceptable responses [2], leaving gaps in potential risk mitigation solutions. A comprehensive examination of recommendations' statuses since Sweedler's [2] effort is overdue. Resistance to implementing recommendations' proposed risk mitigations inhibits improvements, and investigators and investigative bodies must be skilled at strategically overcoming such resistance [10]. Illuminating successful and constrained risk mitigations may empower investigative bodies with additional information, potentially improving their abilities to effect safety enhancements.

Beyond the binary implementation or non-implementation of safety recommendations, the timeliness of recommendations' implementations is also imperative to prevent subsequent occurrences. Karanikas [11] identified the absence of standardized timeliness expectations and associated assessments of safety recommendations as a critical inhibitor of effective safety assurance processes. Karanikas [11] proposed a metric to evaluate the differences between recommendations' due dates and implementation dates. Investigative bodies may, however, lack the authority to enact a due date. Nonetheless, an improved understanding of timeliness from a recommendation's issuance through to closure may provide additional insights into the characterization of risk mitigation strategies.

1.3.2. Aviation Safety Research Employing Qualitative Data Analysis

Velazquez [12] employed the NVivo qualitative data analysis software to explore the presence of 12 behavioral traps across 34 NTSB accident reports. Velazquez [12] reported descriptive results and visualizations to illuminate the presence of behavioral traps in accident report data. Although inter-rater reliability was discussed within the methodological description, the associated results were not reported. Nonetheless, Velazquez's [12] exploration supports this study's use of the NVivo qualitative data analysis software for aviation safety research.

Insley and Turkoglu [13] employed NVivo to explore aircraft maintenance-related accidents' (*n* = 112) causal and contributing factors between 2003 and 2017. They developed and validated a new coding hierarchy, the Maintenance Factors Analysis and Classification System (MxFACS), to elucidate the prevailing maintenance factors among the accident report data. Insley and Turkoglu [13] reported the results found among key case attributes, including geographic regions and accident event categories. While their scope was focused on factual and causal accident report data, in contrast to the current study's focus on recommendations, it effectively demonstrates the utility of NVivo to support qualitative analyses.

1.3.3. Communications and Sentiment Analysis

NTSB accident reports have been subjected to forms of communication analyses. Coogan [14] assessed three rhetorical approaches (materialist, classical, and constructivist) in the context of two NTSB railway accident reports. The NTSB railway accident reports inadequately persuaded the Chicago Transit Authority, who rejected the NTSB recommendations, to address problematic rail communication policies and its associated technologies [14]. Coogan analyzed the NTSB's accident report communications, offering a perspective on improving the persuasive power among rhetoric styles. Coogan's scope included railway accident reports, including their factual information, analysis, conclusions, and recommendations [14]; however, the recommendations' subsequent correspondence between the NTSB and the Chicago Transit Authority may have more effectively illuminated the rhetoric inadequacies at play. While rhetoric and persuasiveness characteristics are not the focus of the current study, Coogan's study provides perspective on the importance of examining recommendations' correspondences.

Orabi [15] analyzed NTSB aviation accident reports (*n* = 76) published between 2000 and 2021, employing a corpus-based genre approach to explore the corpus's text, genre, and professional culture. Orabi [15] found the reports' recommendations to be among the shortest content sections. Nonetheless, the recommendation sections supported the key professional culture corpus attribute of embracing change and learning from mistakes [15]. Orabi's findings reaffirm that NTSB aircraft accident reports are composed using technical, objective, and analytical written communication styles; these styles often mask humanistic and affective elements [16]. In contrast, recommendation correspondence between the NTSB and addressees appears to have affective and sometimes argumentative elements

within its written communications. However, the current literature indicates that a systematic analysis of correspondence sentiments has yet to be performed; this research addresses this gap and contributes to communication analyses by examining sentiment as a critical factor in stakeholder engagement.

1.3.4. Classification Framework

The SHELL model was developed by Edwards in 1972 and evolved by Hawkins in 1975 [6]. The SHELL model provides a conceptual model of the key attributes and interfaces in aviation systems and has been widely accepted in the aviation domain [6]. Summarized by Roggow and Zarei [17] (p. 154), the attributes include "S—Software: the policies, procedures, and regulations; H—Hardware: the machines, equipment, and physical resources; E—Environment: the workspace and ambient conditions; L(x2)—Liveware: the humans, both direct operators central to the model and those interacting with them". Although the SHELL model is often applied retrospectively, such as in accident investigations [17], it can additionally be leveraged to categorize the nature of risk control and mitigation efforts. As such, the SHELL model is a practical framework to inform qualitative data analysis frameworks and node hierarchies.

ICAO, the global authority for civil aviation standardization and harmonization, developed the Accident and Incident Reporting (ADREP) taxonomy to standardize definitions and descriptions for accident and incident reporting, including explanatory factors adapted from the SHELL model [5]. The ADREP explanatory factors include hundreds of attributes spanning the categories of SHELL, with up to five branches of classification depth to support nuanced differences. While this taxonomy's practical use by ICAO Member States is assumed, the extant literature has yet to demonstrate its use as a qualitative data coding hierarchy. Nevertheless, the researcher selected this framework due to its global recognition as a standardized classification scheme [5]. The subsequent section illuminates its adaptation for the study.

This section briefly illuminates extant research on aviation safety recommendations. A systematic qualitative data analysis approach has not been applied to aviation safety recommendation content, and correspondences between the NTSB and addressees have not yet been analyzed; this study closes this gap.

## 2. Materials and Methods

This section contains the methodologies employed in the research. This section's content includes a discussion of the research design, data source, collection process, dataset description, node hierarchy and case classifications, and data analysis approach.

### 2.1. Research Design

The study employed a non-experimental research design and qualitative methods. The study systematically explored the United States' NTSB aviation safety recommendations' textual content to identify the prevailing risk mitigation themes and to characterize the sentiment of the interactions between the NTSB and addressees. Content and sentiment analysis approaches explored the complex themes within recommendations and addressee correspondences, respectively; these approaches provide richer insights than quantitative methods [18]. Qualitative research methods were appropriate because the research intended to explore textual and thematic content to reveal complex interactions and sentiments [18–20].

While content analysis is not itself a novel approach, its application to NTSB aviation safety recommendations has been limited. Additionally, while media sources have scrutinized the interactions between the NTSB and addressees, systematic sentiment analyses of the iterative interactions between the NTSB and addressees have not been employed.

Sentiment analysis is an appropriate means for assessing the degree of neutral objectivity or polarity in communications between parties [20]. The combined use of content analysis and sentiment analysis blends manifest and latent analysis approaches. Through content analysis (manifest), this research addressed what the involved parties had conveyed, using the literal words in the text [21]. Through sentiment analysis (latent), the research sought the underlying feelings and emotions from the text. Together, these approaches were best suited to address the research purpose.

*2.2. Data Source, Collection Process, and Dataset Description*

This research utilized aviation safety recommendation data from the NTSB CAROL database [1]. These data are archived on a publicly accessible government website, and their use is unrestricted. The author employed a CAROL custom query of recommendations with the following rules:

- Safety recommendations/status/date issued is on or after 1 January 2015;
- Safety recommendations/status/date issued is on or before 31 December 2019;
- Safety recommendations/other/recommendation mode is aviation.

The extracted dataset included 187 aviation safety recommendations. Each recommendation included a unique identification number, an issuance date, a priority level, a safety recommendation code(s), a Notice of Proposed Rulemaking (NPRM) designator, an addressee(s), a text body, a status, and a closure date, if applicable. Each recommendation also included a textual accident synopsis; these were not within the scope of the current research. Included with each recommendation were addressee details and dated correspondences between the NTSB and the addressee; these correspondences ranged from two to several dozen per recommendation.

*2.3. Node Hierarchy and Case Classifications*

The author developed an NVivo node hierarchy based on expertise and the extant literature [7]. ICAO, the global authority for civil aviation standardization and harmonization, developed the ADREP taxonomy to standardize definitions and descriptions for accident and incident reporting, including explanatory factors adapted from the SHELL model [5]. The author adopted the first- and second-level categories to support the classification of recommendations' risk mitigation themes. Although ICAO's ADREP taxonomy supports up to five classification levels for a refined specificity, the researcher elected a two-level adaption to balance coding practicality and specificity for the current effort. The researcher directly adopted the first- and second-level categories; Table A1 illustrates the ADREP SHELL attributes' and interfaces' definitions. Node hierarchy code definitions improve the researchers' stability, accuracy, and reproducibility, thereby improving validity and reliability [18,22]. An external subject matter expert reviewed the node hierarchy structure prior to its application. The subject matter expert has formally participated in over one hundred aircraft accident investigations. This external expert's review supports the construct validity of the instrument and its results [7].

The NVivo software, version 14, includes a feature to automatically detect and code sentiment using its word sentiment repository [23]. NVivo's word sentiment repository includes pre-defined assignments of very negative, moderately negative, neutral, moderately positive, and very positive. However, the sentiment coding is not able to recognize or contextualize sarcasm, double negatives, slang, dialect variations, idioms, or ambiguity [23]. Therefore, a user-defined NVivo node hierarchy was not needed for the sentiment analysis, but a careful review of the sentiment coding results was needed and is described in the next section.

NVivo case classifications supported comparative analyses of recommendation file cases among categorical attributes. The case classification attributes employed in this study included the recommendation identification number, year, status, addressee, priority level, times reiterated, NPRM, days open, and correspondence count.

### 2.4. Data Analysis Approach

The research utilized the NVivo qualitative data analysis software, version 14, to explore, code, and analyze the qualitative data. Woods et al. [24], from systematic analyses of qualitative data analysis software (QDAS) publications, emphasized that researchers must immerse themselves in their data and avoid over-reliance on software's automated processes; this advice was heeded throughout the research. The author interactively coded the recommendations' content using the node hierarchy in Table A1. The node hierarchy and coding definitions were referenced in each coding session to sustain stability, accuracy, and reproducibility [22].

NVivo's sentiment auto-coding was employed to analyze the correspondences between the NTSB and addressees, enabling the sentiment characterization of interactions in addressing aviation recommendations. The sentiment auto-coding function employed a repository of pre-defined scores (very negative, moderately negative, neutral, moderately positive, and very positive) for words [23]. The sentiment auto-coding assessed the sentiment of words in isolation and may be unable to contextualize the sentiment. Therefore, the author reviewed each case's sentiment auto-coding results and modified the coding, as summarized as follows:

- Sentiment auto-coding was initially applied to all segments of the recommendation and correspondence case files. In all cases, the researcher modified the auto-coding to ensure that its application was solely to the correspondence content.
- The NTSB sporadically included multiple recommendations in the content of their initial correspondence to the addressee. In these cases, the sentiment auto-coding was removed to ensure that results were not duplicated across cases.
- The sentiment auto-coding was unable to successfully contextualize some negative words, e.g., failure, when they were objectively used, such as an aircraft system failure. In such cases, the researcher reassigned a neutral sentiment.

Descriptive statistics and visualizations effectively communicate qualitative data analysis results [7,22,25]. In addition to descriptive statistics and visualizations, NVivo's crosstab coding query quantified each recommendation's coding counts across all parent and child nodes. The crosstab coding results revealed the prevailing risk mitigation themes among the NTSB aviation recommendations.

## 3. Results

This section presents the data exploration, visualization, and analysis results. The section begins with descriptive statistics; thereafter, the NTSB aviation recommendations' prevailing risk mitigation themes and the sentiment analyses of the NTSB and addressee interactions are presented.

### 3.1. Descriptive Statistics

The NTSB issued 187 aviation safety recommendations between 1 January 2015 and 31 December 2019. As shown in Figure 1, 2016 and 2017 included the most recommendations, at 54 and 45, respectively, whereas 2015, 2018, and 2019 had comparably fewer. Among all recommendations, the majority resulted from aircraft accident and incident investigations. A safety research study on improving pilot weather report submission and dissemination ($n = 19$) and an NTSB emerging flight data and locator technology forum ($n = 8$) resulted in 27 recommendations.

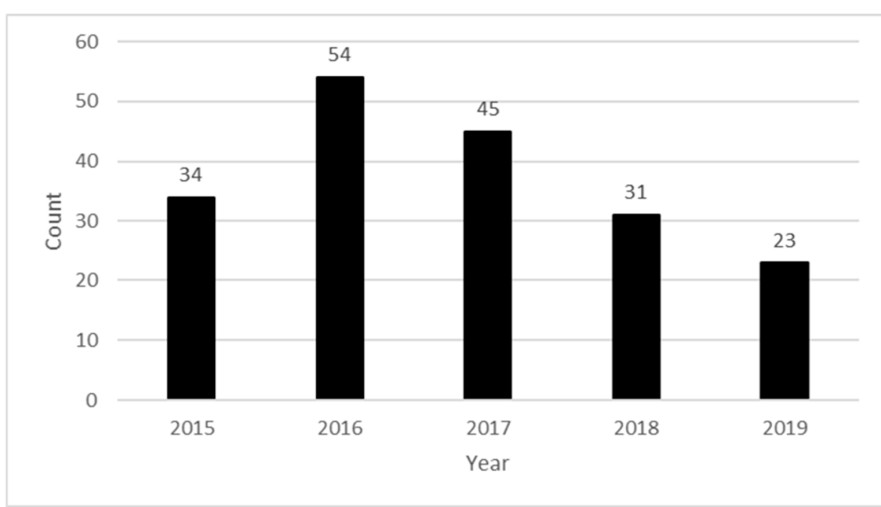

**Figure 1.** Recommendations issued per calendar year.

The NTSB's [1] addressee categories include associations, federal government, foreign, local government, private industry, state government, and union. The federal government category most typically consists of the FAA, but can also include other federal government agencies, such as the National Oceanic and Atmospheric Administration. Further, private industry includes both original equipment manufacturers (OEMs) and operators, e.g., an airline. Hence, the author elected an alternative scheme to improve the categorical discrimination for this aviation-focused project. As per Figure 2, the FAA was the prevailing addressee ($n = 131$; 70.1%); however, foreign equivalents, generically labeled foreign civil aviation authority (CAA), also received four recommendations. Five non-FAA U.S. government agencies were recommendation addressees. Among private industry recommendation addressees, most were OEMs ($n = 19$), with just three instances of operators. Associations, including industry, trade, technical, and labor, were addressed 15 times. In ten instances, multiple entities were addressed in a single recommendation; most often, these were multiple associations.

The status of the 187 aviation safety recommendations, as of 31 January 2025, is illustrated in Figure 3. In total, 69 recommendations (36.9%) remain open; of these open recommendations, 52 were classified with an acceptable response, 3 were classified with an acceptable alternate response, and 14 were classified with an unacceptable response. There were 118 (63.1%) total closed recommendations, including 68 acceptable actions, 11 acceptable alternative actions, and a single instance of an exceeded recommended action. In four instances, the NTSB reconsidered its recommendation; this classification was typically associated with the addressee having completed risk mitigation action in the weeks preceding the recommendation's formal issuance. In five instances, the NTSB closed its recommendation as no longer applicable because the addressee ceased to exist as an organization or association. However, 29 recommendations were closed with unacceptable action; in 3 cases, the closure was superseded by a subsequent accident or incident recommendation associated with the same unresolved risk mitigation.

The dataset included 15 recommendations that were reiterations of previously issued recommendations. In total, 10 of these 15 recommendations were first-time reiterations; 2 were second-time reiterations. Recommendation A-16-036 was reiterated for a seventh time; this recommendation was successfully closed on 9 August 2024, reflecting the 2024 Final Rule to expand Safety Management System (SMS) requirements to 14 CFR Part 135 operators. However, recommendations A-16-035, a sixth reiteration to the FAA for Part 135 flight data monitoring programs, and A-16-034, a fifth reiteration to the FAA to require the installation of flight data recording devices in support of the former, remain open with unacceptable responses.

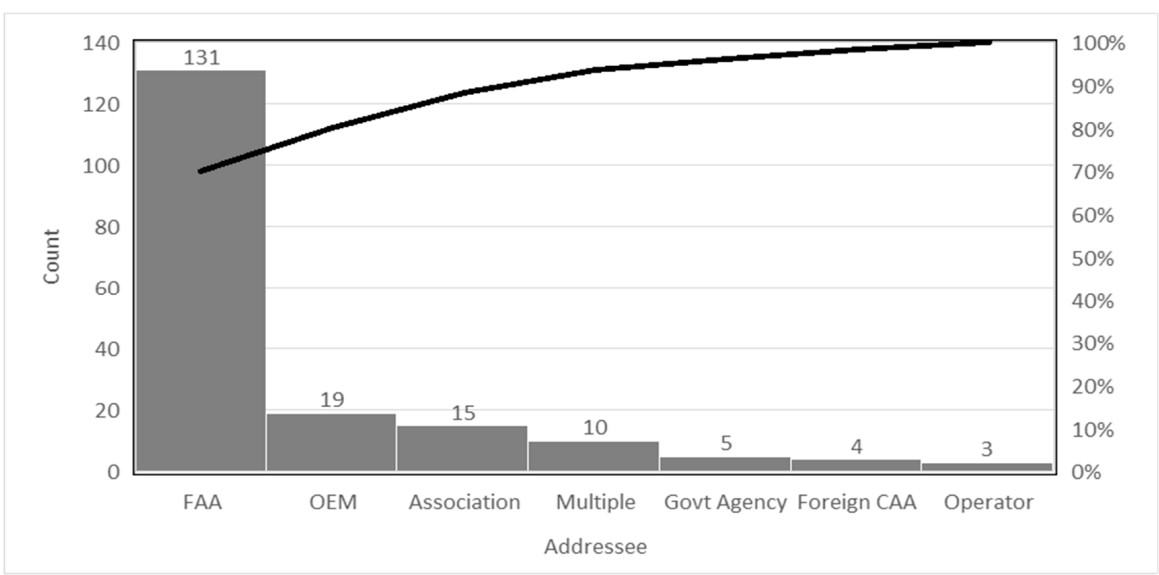

**Figure 2.** Recommendation distributions among addressee categories. The count is displayed on the left-side y-axis and the percentage of the total is displayed on the right-side y-axis. Pareto line associates with the percentage.

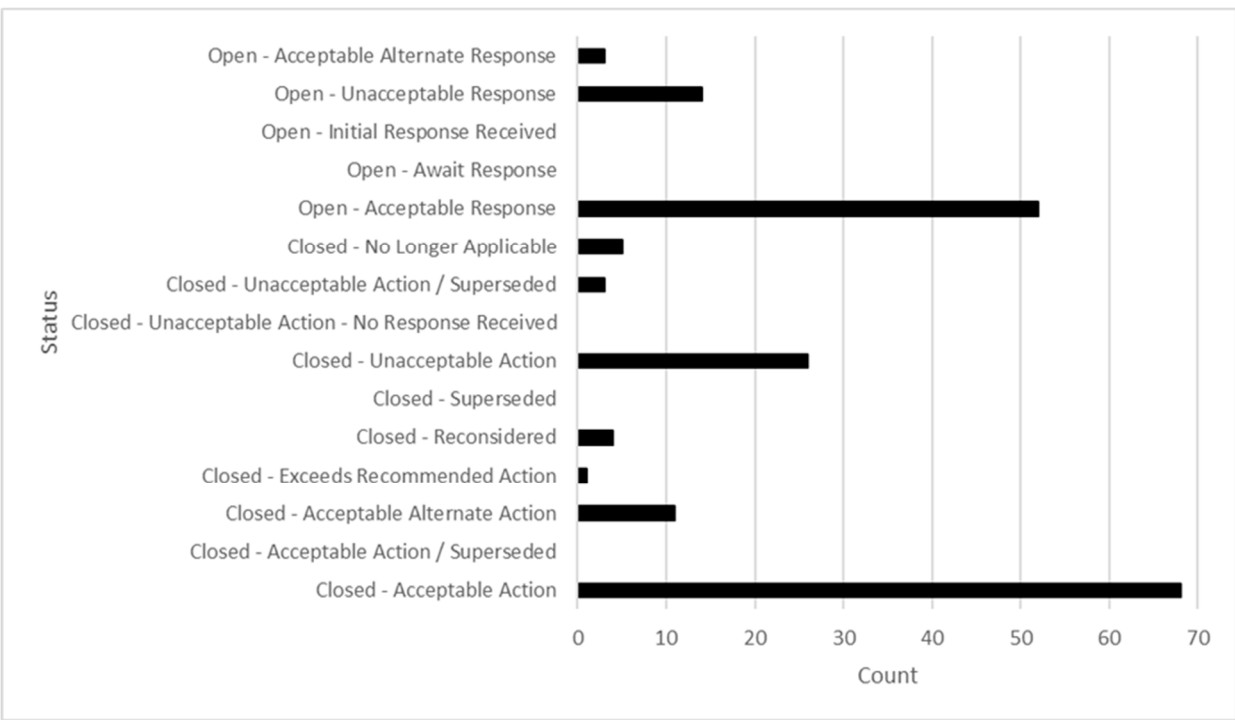

**Figure 3.** Recommendation statuses distribution, using all NTSB status categories.

The recommendations' durations from issuance to closure and total correspondence counts notably varied across the dataset; Table 1 illustrates these results. The correspondence count was measured for all recommendation statuses. It was noted that this dataset included 69 open reports as of 31 January 2025. Therefore, the correspondence mean, median, and other values may increase. On the other hand, the duration values included only recommendations that were closed. Open recommendations spanned from 2015 to the end of 2019, with steadily increasing durations. These recommendations, upon closure, will significantly increase the magnitude of duration, as they were open for more than a

thousand days and counting. Beyond descriptive statistics, these variables were employed in contextualizing the sentiment analyses and are revisited later.

**Table 1.** Descriptive statistics: days between the recommendation issuance to closure (all closed recommendations) and correspondence counts (all recommendations).

| Variable | Min. | Max. | Mean | *SD* | Median |
|---|---|---|---|---|---|
| Duration (days) | 84 | 2836 | 1381.4 | 730.3 | 1463 |
| Correspondence (count) | 2 | 22 | 7.3 | 3.5 | 7 |

The dataset included primarily non-urgent ($n = 185$) priority-level recommendations. The NTSB [1] issues urgent safety recommendations when there is a pressing safety issue, and they expect the addressee to take immediate action. Urgent recommendation A-17-001 asked the FAA to issue an emergency airworthiness directive (AD) that required the owners and operators of Piper PA-31T-series airplanes to take actions that address potentially unsafe wiring conditions. Recommendation A-17-001 was closed in 107 days, though the FAA AD was issued over two months earlier. Urgent recommendation A-18-012 asked the FAA to prohibit all open-door commercial passenger-carrying aircraft flights that use passenger harness systems unless the system allows passengers to rapidly release the harness with minimal difficulty and without having to cut or forcefully remove the harness. This recommendation was closed in 129 days; however, the FAA issued the prohibition just three days after the recommendation was issued.

The dataset included 20 recommendations classified as *yes* for Notice of Proposed Rule-making (NPRM). However, the author identified that this classification was inconsistently assigned as *no*, even with contrary evidence in the correspondence. A subsequent results section includes an alternative approach to revealing rulemaking and regulatory efforts.

Lastly, a word cloud illustrates the most frequently occurring words in the largest font sizes, with incrementally smaller font sizes representing less frequently occurring words [7]. Figure 4 represents the 100 most common words across all 187 recommendations and their associated correspondences. The word cloud shows the visual density and distribution of the qualitative dataset.

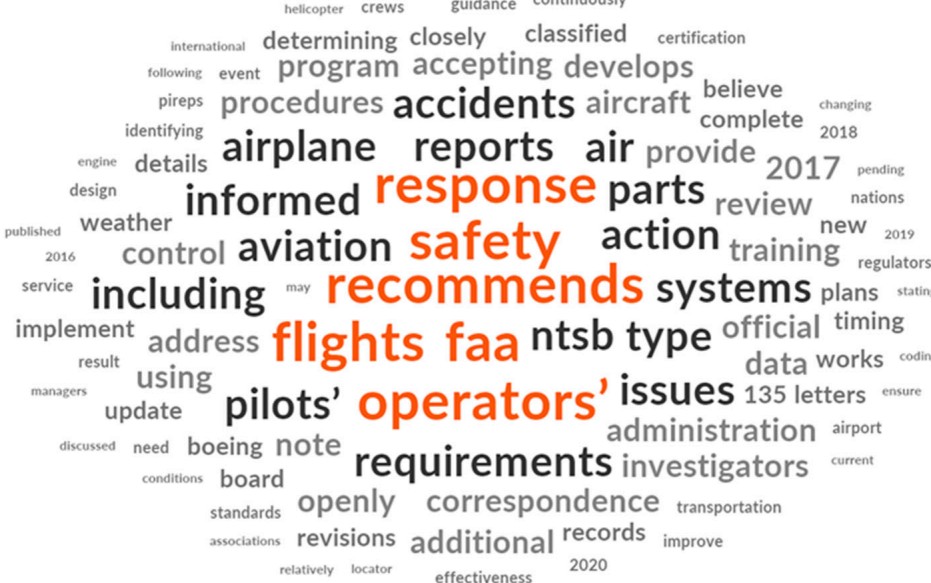

**Figure 4.** NVivo word cloud: all recommendations and correspondence. The largest and central words represent those occurring most frequently across all recommendations and correspondences.

*3.2. Aviation Recommendations—Risk Mitigation Themes*

The author explored the 187 aviation safety recommendations' prevailing risk mitigation content themes by interactively coding their textual content with the node hierarchy illustrated in Table A1. The results are visually illustrated using an NVivo node hierarchy graphic in Figure A1; the node size differences represent the assigned codes' proportionality. Alternatively, Table A2 includes the numeric distribution of the risk mitigation themes.

More than half (56.7%) of the recommendations included an environment-liveware (E-L) attribute related to company, management, manning, or regulatory issues. Recommendations frequently included a proposed improvement to FAA regulations or guidance or organizational policies and practices; the proposed regulatory or administrative improvements were often crafted to more systematically improve the regulatory or operational environment, thereby minimizing subsequent deficiencies. Relatedly, liveware-liveware (L-L) regulatory activities (25.1%) illuminated the risk mitigation actions that directly influence how people interface with regulatory standards.

The physical environment was the next most frequently categorized risk mitigation theme. There were numerous instances related to improved weather information accuracy, reporting, dissemination, communication, and displays; additionally, there were several physical environment mitigation themes related to interfaces with other natural environment attributes, e.g., darkness or terrain, and human-made attributes, e.g., airport layouts.

Software-liveware (S-L) risk mitigation categories were also quite frequent. Other human-system support issues were found in 31% of all cases and were most often related to developing and distributing best practices or technical documents intended to influence human performance. Procedures (26.7%) and training (20.9%) were also common themes, and target groups spanned the aviation industry, including pilots, instructors, air traffic controllers, maintainers, load planners, and FAA inspectors. The next three subsections illustrate how these risk mitigation themes varied based on addressees, statuses, and reiterations.

3.2.1. Risk Mitigation Themes—Addressee

Analyzing the differences in risk mitigation among the addressee categories provides an interesting opportunity for comparison, both within and among addressees, as illustrated in Figure A2. The risk mitigation themes of recommendations addressed to the FAA most often included the E-L category of company, management, manning, or regulatory issues. Thereafter, the following three risk mitigation categories were the next most common: E-L physical environment, S-L other human-system support issues, and L-L regulatory activities. While it may seem intuitive that L-L regulatory activities would be associated with E-L company, management, manning, or regulatory issues, L-L regulatory activities were only classified when the FAA-addressed recommendation provided evidence of a regulatory interface at the L-L level. Further, there were several instances of recommended improvements in organization, management, or manning for the FAA itself.

OEMs, e.g., Boeing, Airbus, and Bombardier, were the next most common addressees. In contrast to the FAA, very few recommendations addressed the E-L category of company, management, manning, or regulatory issues. Instead, the risk mitigation recommendations addressed to OEMs were most typically associated with hardware-liveware (H-L) interfaces, followed by S-L procedures. The OEM-directed risk mitigation recommendations occasionally included H-L information and data sources, S-L training, and the related categories of both E-L operational task demands and liveware's personal workload management. These findings were not unexpected given OEMs' purview of system and component design and the associated effects on procedures, training programs, tasks, and workload. There were

also less frequently coded but nonetheless important categories, including H-L automatic defenses or warnings and human-software interfaces.

The third-most frequent addressee was associations; further, many of the multiple addressee classifications included multiple associations. The most prevalent risk mitigation theme for associations was E-L physical environment, followed by E-L company, management, manning, or regulatory issues, S-L procedures, and S-L system support issues. In several cases, associations were solicited to encourage their membership or constituents to review or improve latent factors within their organization, including policies, procedures, or guidance; these facets were often related to operational interfaces with the physical environment, such as weather or terrain.

In other instances of multiple addressees, multiple training providers or associations affiliated with flight training were asked to either directly enhance S-L training or encourage their constituents to do so. Hence, S-L training and liveware experience, knowledge, and recency were prevalent themes. In two peculiar instances, the FAA and three contract air traffic organizations were asked to include recent midair collision examples in initial and recurrent training for controllers (recommendation A-16-051), as well as brief their supervisory and frontline personnel on air traffic control errors in the accidents (recommendation A-16-052). While the contract air traffic organizations satisfactorily addressed these recommendations, the NTSB deemed that the FAA's adherence to their recommendations was unacceptable.

### 3.2.2. Risk Mitigation Themes—Status

The differences in risk mitigation themes among the recommendations' statuses revealed a few interesting observations. However, the author emphasizes merely observations, as statistical comparisons were beyond the scope of the current study. Table A3 includes the coded risk mitigation themes across the following four refined status categories: Open—All Acceptable, Closed—All Acceptable, Open—All Unacceptable, and Closed—All Unacceptable. The reconsidered and no longer applicable status categories were of minimal comparative value, so they were excluded from this table. Both open and closed acceptable statuses included their respective acceptable alternate action and exceeded recommended action categories. Similarly, the closed unacceptable status included superseded.

Visual inspection and comparison among these values reveal balanced distributions in most risk mitigation categories. As reflected earlier, the E-L category of company, management, manning, or regulatory issues was prevalent across all recommendations; however, its frequency of association with unacceptable statuses was relatively higher than that with acceptable statuses. Similarly, the liveware category of experience, knowledge, and recency and the S-L training category appeared to be more frequently associated with unacceptable statuses. In contrast, all recommendations with human physiology risk mitigation themes associated with acceptable statuses, and the remaining liveware categories favored acceptable statuses, except for the aforementioned experience, knowledge, and recency category.

### 3.2.3. Risk Mitigation Themes—Reiteration

Analyzing the differences in risk mitigation categories for reiterated recommendations provides some interesting comparisons, as illustrated in Table A4. Again, the author emphasizes that statistical tests for differences were beyond the scope of this qualitative, exploratory study. Among the 187 recommendations, 10 were reiterated once. Since relatively few recommendations were reiterated two or more times ($n = 5$), these were binned into a single, aptly named category for this section's results.

First, it was observed that liveware-centered risk mitigation actions rarely need reiteration. At least one child node category appeared noteworthy among each interfacing parent node category. Reiterated recommendations included a disproportionately higher prevalence of the following risk mitigation categories: S-L other human-system support, H-L inadequate information or data sources, L-L regulatory activities, and the E-L category of company, management, manning, or regulatory issues.

### 3.3. NTSB and Addressee Correspondence—Sentiment Analyses

NVivo sentiment auto-coding was applied to all correspondence between the NTSB and recommendation addressees. The sentiment analyses' results were reviewed to ensure that NVivo's repository of word characterizations was appropriately contextualized. The following subsections explore the results by addressee, status, time, reiteration, and correspondence counts.

### 3.3.1. Sentiment Analyses—Addressee

The recommendation correspondences' sentiment distributions among addressee categories are illustrated in Figure A3. Correspondence with government agencies, excluding the FAA, was observed to be distributed with an overall neutral sentiment compared to other addressees. Correspondences with government agencies were very nearly balanced between positive and negative sentiment content. While only five recommendations were issued to government agencies, addressee categories with just three (operator) or four (foreign CAA) recommendations were more aligned with all other addressees, indicating a potential uniqueness of the neutral-sentiment correspondence with government agencies.

Aside from the government agency category, all other addressee correspondence sentiment analyses favored a slightly positive imbalance. As per Figure A3, all other addressees' very positive sentiment results were reasonably balanced with the very negative results; however, the moderately positive sentiment outweighed the moderately negative sentiment, in many cases by an increment of 15% or more. Further, the addressee categories of *multiple* and *associations* demonstrated a greater observable imbalance toward positive sentiment. Given the earlier assessment that many *multiple* addressee recommendations included several *associations*, a theme of greater positivity sentiment appeared to be particularly affiliated with *associations*.

### 3.3.2. Sentiment Analyses—Status

The recommendation correspondences' sentiment distributions were examined across the following four refined status categories: Open—All Acceptable, Closed—All Acceptable, Open—All Unacceptable, and Closed—All Unacceptable (Figure A4). The reconsidered and no longer applicable status categories were excluded from this figure. Open and closed acceptable statuses included their respective acceptable alternate action and exceeds recommended action categories. Similarly, the closed unacceptable status included superseded.

First, while very positive sentiments did not appreciably differ among all statuses, unacceptable statuses, whether open or closed, included approximately twice as many very negative sentiments as acceptable statuses; these results reinforce the author's observation of strongly crafted correspondences when recommendations failed to be adequately addressed. Interestingly, the overall sentiment distribution between acceptably closed recommendations was very similar to that of open recommendations with unacceptable responses, approximately 60% positive. However, these statuses differed through stronger sentiments within the negative polarity for open recommendations with unacceptable responses compared to more moderately negative sentiments for acceptably closed recommendations.

3.3.3. Sentiment Analyses—Time

The recommendation correspondences' sentiment distributions among the calendar years of issuance are illustrated in Figure A5. The author emphasizes that these distributions are based on the recommendation issuance year, not the year of each individual correspondence. The calendar years from 2015 to 2018 were observed to have very similar sentiment distributions, leaning toward slightly positive. The calendar year of 2019, however, was observed to have relatively less negative sentiment coding, contributing to a higher percentage of moderately positive sentiment. Across the 23 recommendations issued in 2019, all but 2 included acceptable status classifications, potentially influencing these results.

The recommendation correspondences' sentiment distributions were also analyzed based on the number of days between recommendation issuance and closure. Given that open recommendations do not yet have a closure date, the 69 open recommendations were excluded from this analysis. Figure A6 illustrates the sentiment distribution among the 118 closed recommendations, sorted into three nearly equal groups. The slightly positive sentiment distribution was nearly consistent among reports with short, medium, and long durations until closure. Subtly, it appeared that medium durations, from more than 958 days to 1766 days, included slightly fewer extreme positive and negative sentiment values.

3.3.4. Sentiment Analyses—Reiteration

The recommendation correspondences' sentiment distributions based on reiterations are illustrated in Figure A7. The dataset included 15 recommendations that were reiterated one or more times. The sentiment analyses for these recommendations' correspondences differed subtly from those not needing reiterations. Although the overall results between non-reiterated and reiterated recommendations both favored slightly positive, reiterated recommendation correspondence included distributions with higher frequencies of strong sentiment, both positive and negative. These results may indicate increased gratification for closure and frustration about continued gaps with recommendations for unaddressed, reiterated mitigation actions.

3.3.5. Sentiment Analyses—Correspondence Counts

Lastly, the recommendation correspondences' sentiment distributions based on correspondence count are illustrated in Figure A8. The author distributed the correspondence counts into quartiles for visualization and comparison. No noteworthy patterns were discernable; however, this finding is intriguing. These results may indicate that the sentiments remained quite consistent and slightly positive even as correspondences between the NTSB and addressees increased.

## 4. Discussion and Conclusions

This research qualitatively explored 187 NTSB aviation safety recommendations issued between 2015 and 2019, inclusively. The research utilized an adapted ICAO SHELL-based ADREP framework to elucidate the prevailing risk mitigation themes among the recommendations' content. The research also utilized NVivo auto-coding features to explore the sentiment of correspondences between the NTSB and addressees.

The FAA was the most frequent addressee of NTSB aviation safety recommendations ($n$ = 131; 70.1%). Across all recommendations, 69 (36.9%) remain open, and just 14 open recommendations were classified with an unacceptable response; 118 (63.1%) recommendations were closed, mostly through acceptable actions ($n$ = 68) or acceptable alternative actions ($n$ = 11). However, 29 (15.5%) recommendations were closed with unacceptable action; among these, the majority included a proposed risk mitigation approach related to

company, management, manning, or regulatory issues. A subsequent examination of these recommendations is warranted to identify the key factors that inhibited their acceptance, such as institutional conflicts of interest or excessive implementation or compliance costs.

Reiterated recommendations included a disproportionately higher prevalence of the following risk mitigation categories: software–liveware other human-system support, hardware-liveware inadequate information or data sources, liveware-liveware regulatory activities, and environment-liveware company, management, manning, or regulatory issues. However, the small percentage ($n = 15$; 8%) of reiterated recommendations in this dataset warrants further investigation. The exploration of a broader dataset of reiterated recommendations, e.g., 2000–2019, could more deeply examine risk mitigation themes that have been inadequately addressed. Additionally, sentiment analyses of such a dataset could reveal specific negative themes among addressees, illuminating areas for additional intervention, such as congressional action.

Most recommendations (56.7%) employed a risk mitigation approach related to company, management, manning, or regulatory issues. While insightful, this risk mitigation category included several approaches; further research could examine this dataset segment to more deeply assess the subcategories within this risk mitigation theme. Furthermore, a subsequent effort could further analyze this dataset segment to examine the utility of regulatory changes, similar to Waycaster et al.'s [9] approach. While Waycaster et al. examined all regulatory changes between 2002 and 2009, an approach specific to NTSB aviation safety recommendations could quantify the rulemaking costs and benefits stemming from these recommendations.

The author also noted the peculiar absence of a key theme across the dataset. The NTSB's MWLs in 2015, 2016, and 2017/2018 included the objective to "prevent loss of control in flight in general aviation" [3]. The Aircraft Owners and Pilots Association [26] reported that loss of control (LOC) continued as a frequent causal factor of aircraft accidents through 2019. LOC was not cited among any of the 187 aviation safety recommendations. Two aviation safety recommendations, A-16-012 and A-16-013, included correspondence that justified the recommendations' capabilities to reduce LOC accidents, amongst other causal factors. Nonetheless, the scarcity of LOC recommendations within the dataset was disproportionate to its frequency as a causal factor and its continued presence on NTSB MWLs. Although the MWL has been retired, this finding illuminates the importance of ensuring that spotlighted safety issues are justified by recommendations.

The sentiment analyses resulted in several interesting observations. First, correspondences with government agencies other than the FAA were distributed with an overall neutral sentiment compared to the slightly positive imbalance for most other addressees; however, further analyses with a broader dataset of government agencies should be used to validate this observation. A theme of higher positivity was particularly evident within associations' correspondences. While most associations were safety allies for NTSB recommendation deployment, there were also instances in which associations never responded to NTSB requests, e.g., A-17-015. Second, unacceptable statuses, either open or closed, included about twice as many very negative sentiments as acceptable statuses. Stronger sentiments of disappointment or frustration may not have been able to sufficiently influence or pressure addressees into acceptable action, commensurate with Coogan's [14] findings. Third, reiterated recommendations' correspondences included higher frequencies of strong sentiment, both positive and negative. Fourth, minimal sentiment variations were observed based on either time and correspondence counts, indicating that sentiment remained relatively consistent and slightly positive between the NTSB and addressees, even as time passed and interactions increased. The study's sentiment analyses findings also suggest opportunities to further enhance recommendation correspondence phrasing with styles

that have garnered acceptance from addressee stakeholders. Overall, improved sentiment awareness may enable reflective efforts to shape and enhance stakeholder relations and their subsequent interactions.

In addition to the future research possibilities already noted in this section, a subsequent iteration of this study could examine risk mitigation themes within a revised timeframe. The current study's timeframe was selected to ensure that all accident investigations had been completed. Given that NTSB aircraft accident investigations are typically completed within one to two years after the occurrence, a timeframe from 2020 to 2023 could support the next iteration of this study. Further, this timeframe could reveal nuances related to the COVID-19 pandemic. However, this approach would likely need to be decoupled from the sentiment analyses. As noted in the sentiment analyses by time, this study's dataset included 69 open recommendations; these associated correspondences will evolve through their closure. Hence, future sentiment analyses should consider datasets with increased proportions of closed recommendations. Although the practical relevance may diminish by using older recommendation correspondence data, the comprehensiveness and generalizability of the sentiment analyses would improve.

Furthermore, this study focused on NTSB aviation safety recommendations, which primarily involved accidents occurring within the United States. Given the global nature of the aviation industry, future research could explore and compare risk mitigation themes among additional investigative bodies' recommendations; such an effort could reveal risk mitigation themes that are specific to geographic areas or regions. However, the approach to sentiment analyses employed by this study may not be conducive at a global or regional scale, given the potential inhibitors related to correspondence information accessibility.

Lastly, a subsequent study could analyze risk mitigation themes and correspondences among other modalities. The NTSB also develops safety recommendations across highway, marine, pipeline, railway, and multimodal transportation. Examinations of safety recommendations' risk mitigation themes among other modes could better illuminate the risks that are shared among modes compared to those unique to aviation. Further, sentiment analyses of addressee correspondences across additional modes could similarly elucidate similarities and differences.

## 5. Limitations

This study is not without limitations. The study employed a single researcher to qualitatively code the data. Although an external expert was used to help review and validate the adopted ICAO ADREP node hierarchy, the study assumed that the instrument's original design was validated. Multiple researcher coders may have better demonstrated the reliability of the node hierarchy's application through the assessment of inter-rater reliability. Additionally, this study employed descriptive statistics and data visualizations to explore and observe patterns within the qualitative data. Statistical tests were not employed to quantitatively assess relationships; although this was beyond the scope of the current study, a subsequent examination of this dataset could address this limitation.

**Funding:** This research received no external funding.

**Institutional Review Board Statement:** Not applicable.

**Informed Consent Statement:** Not applicable.

**Data Availability Statement:** The data presented in this study are available in NTSB's CAROL at https://data.ntsb.gov/carol-main-public/basic-search (accessed on 31 January 2025). These data were derived using the process described in Section 2.2 of this article.

**Acknowledgments:** While the study was not directly funded, the author completed this study to support his scholarly activity at Embry-Riddle Aeronautical University (ERAU); hence, ERAU is acknowledged for its project support.

**Conflicts of Interest:** The author declares no conflicts of interest. The author has no known conflicts of interest with the NTSB or addressees or the nature of the safety issues and recommendations.

## Abbreviations

The following abbreviations are used in this manuscript:

| | |
|---|---|
| NTSB | National Transportation Safety Board |
| ICAO | International Civil Aviation Organization |
| CAROL | Case Analysis and Reporting Online |
| MWL | Most Wanted List |
| ADREP | Accident/Incident Data Reporting |
| FAA | Federal Aviation Administration |
| HFACS | Human Factors Analysis and Classification System |
| GPWS | Ground Proximity Warning System |
| MxFACS | Maintenance Factors Analysis and Classification System |
| NPRM | Notice of Proposed Rulemaking |
| QDAS | Qualitative Data Analysis Software |
| OEM | Original Equipment Manufacturer |
| CAA | Civil Aviation Authority |
| SMS | Safety Management System |
| AD | Airworthiness Directive |
| E-L | Environment-liveware |
| L-L | Liveware-liveware |
| S-L | Software–liveware |
| H-L | Hardware-liveware |
| AOPA | Aircraft Owners and Pilots Association |
| LOC | Loss of Control |
| CRM | Crew Resource Management |

## Appendix A

**Table A1.** ICAO ADREP Explanatory Factors (modeled using SHELL): node hierarchy and definitions. Parent node is first level, and child node is second level. Adapted from ICAO [5].

| Parent Node/Child Node | Definition |
|---|---|
| **Software-Liveware (S-L)** | |
| Human interface-procedures | Factors related to the interface between liveware [human] and system support procedures themselves. |
| Human interface-training | Factors related to the interface between liveware [human] and training. |
| Other human-system support issues | Factors related to the interface between liveware [human] and system support issues. |
| **Hardware-Liveware (H-L)** | |
| Human-hardware interface | Factors related to the interface between the human and the system (hardware) interface. |
| Inadequate information/ data sources | Factors related to the liveware-hardware interface associated with the lack of availability of information, inaccurate information, or intermittent information. |
| Human-software/ firmware interface | Factors related to the interface between the human and the system software/firmware interface. |
| Automation/automatic systems | Factors related to the use of automation/automatic systems; this refers to all automatic systems, including automation in ATC control rooms. |

Table A1. *Cont.*

| Parent Node/Child Node | Definition |
|---|---|
| Automatic defenses/warnings | Factors related to the interface between the human and the system automatic defenses/warnings. These keywords should only be used if there is a problem with the warnings, e.g., warnings not available/not working, warnings misleading, or too many false alarms. |
| Operational material | Factors related to the interface between the human and the system which are not covered by hardware or software/firmware (either direct operational or indirect). |
| **Environment-Liveware (E-L)** | |
| Physical environment | Factors related to the interface between the human and the physical environment. |
| Psychosocial factors | Factors related to psychosocial issues associated with or affecting work, e.g., cultural differences. |
| Company, management, manning, or regulatory issues | Factors related to company, management, manning, or regulatory issues which tend to be outside the individual's control and which may affect performance or safety. |
| Operational task demands | Factors related to operational task demands, i.e., demands directly associated with the operational task itself, e.g., flying, navigating, controlling, or servicing part of an aircraft. |
| **Liveware-Liveware (L-L)** | |
| Communications | Factors related to the interface between humans in relation to communications. |
| Team skill/CRM | Factors related to the interface between humans in relation to interactions between people; teamwork. |
| Supervision | Factors related to the interface between humans in relation to supervision. |
| Regulatory activities | Factors related to the interface between humans in relation to regulatory activities. |
| Other human-human interfaces | Factors related to the interface between humans in relation to other human-human interfaces. |
| **Liveware** | |
| Experience, knowledge, and recency | Factors related to experience, qualifications, knowledge, and recency. The keywords should only be used for inexperience, inadequate qualifications, poor knowledge, or lack of recency. |
| Human physiology | Factors related to the physiological conditions of people. Physiology is the science of the normal functions and phenomena of living things. |
| Personal workload management | Factors related to management of one's own or another's workload (if within one's own control). |
| Personal physical or sensory limitations | Factors related to a person's physical or sensory limitations, not including physiological, psychological, or visual illusions. |
| Psychological limitations | Factors related to anything which involves thinking or acting (not including physiological issues), such as learning, memory, personality, or attitudes. |

Table A2. Risk mitigation themes: all recommendations. Frequencies are relative to total recommendations (*n* = 187). Recommendations consistently included two or more coded risk mitigation themes; therefore, percentages exceed 100%.

| Parent Node/Child Node | Coding Count | Frequency (%) |
|---|---|---|
| **Software-Liveware (S-L)** | - | - |
| Human interface-procedures | 50 | 26.7% |
| Human interface-training | 39 | 20.9% |
| Other human-system support issues | 58 | 31.0% |
| **Hardware-Liveware (H-L)** | - | - |
| Human-hardware interface | 36 | 19.3% |
| Inadequate information/data sources | 33 | 17.6% |
| Human-software/firmware interface | 21 | 11.2% |
| Automation/automatic systems | 8 | 4.3% |
| Automatic defenses/warnings | 16 | 8.6% |
| Operational material | 26 | 13.9% |

**Table A2.** *Cont.*

| Parent Node/Child Node | Coding Count | Frequency (%) |
|---|---|---|
| **Environment-Liveware (E-L)** | - | - |
| Physical environment | 62 | 33.2% |
| Psychosocial factors | 2 | 1.1% |
| Company, management, manning, or regulatory issues | 106 | 56.7% |
| Operational task demands | 33 | 17.6% |
| **Liveware-Liveware (L-L)** | - | - |
| Communications | 18 | 9.6% |
| Team skill/CRM | 15 | 8.0% |
| Supervision | 6 | 3.2% |
| Regulatory activities | 47 | 25.1% |
| Other human-human interfaces | 12 | 6.4% |
| **Liveware** | - | - |
| Experience, knowledge, and recency | 40 | 21.4% |
| Human physiology | 9 | 4.8% |
| Personal workload management | 23 | 12.3% |
| Personal physical or sensory limitations | 10 | 5.3% |
| Psychological limitations | 11 | 5.9% |
| **Total** | 681 | - |

**Table A3.** Risk mitigation themes: recommendation status. Excludes reconsidered and no longer applicable statuses. Acceptable statuses include acceptable alternate and exceeded actions. Unacceptable statuses include superseded. Recommendations consistently included two or more coded risk mitigation themes; therefore, percentages exceed 100%. Percentages (%) are specific to each column.

| Parent Node/Child Node | Open—All Acceptable (55) | Closed—All Acceptable (80) | Open—All Unacceptable (14) | Closed—All Unacceptable (29) |
|---|---|---|---|---|
| **Software-Liveware (S-L)** | - | - | - | - |
| Human interface-procedures | 13 (23.6%) | 25 (31.3%) | 2 (14.3%) | 8 (27.6%) |
| Human interface-training | 6 (10.9%) | 16 (20.0%) | 3 (21.4%) | 10 (34.5%) |
| Other human-system support issues | 17 (30.9%) | 25 (31.3%) | 4 (28.6%) | 9 (31.0%) |
| **Hardware-Liveware (H-L)** | - | - | - | - |
| Human-hardware interface | 7 (12.7%) | 21 (26.3%) | 3 (21.4%) | 4 (13.8%) |
| Inadequate information/data sources | 14 (25.5%) | 7 (8.8%) | 5 (35.7%) | 6 (20.7%) |
| Human-software/firmware interface | 7 (12.7%) | 6 (7.5%) | 5 (35.7%) | 3 (10.3%) |
| Automation/automatic systems | 6 (10.9%) | 1 (1.3%) | 1 (7.1%) | 0 |
| Automatic defenses/warnings | 9 (16.4%) | 5 (6.3%) | 1 (7.1%) | 1 (3.4%) |
| Operational material | 15 (27.3%) | 10 (12.5%) | 0 | 1 (3.4%) |
| **Environment-Liveware (E-L)** | - | - | - | - |
| Physical environment | 18 (32.7%) | 28 (35.0%) | 4 (28.6%) | 10 (34.5%) |
| Psychosocial factors | 0 | 0 | 0 | 2 (6.9%) |
| Company, management, manning or regulatory issues | 34 (61.8%) | 37 (46.3%) | 11 (78.6%) | 18 (62.1%) |
| Operational task demands | 11 (20.0%) | 13 (16.3%) | 1 (7.1%) | 7 (24.1%) |
| **Liveware-Liveware (L-L)** | - | - | - | - |
| Communications | 4 (7.3%) | 7 (8.8%) | 0 | 4 (13.8%) |
| Team skill/CRM | 2 (3.6%) | 5 (6.3%) | 2 (14.3%) | 4 (13.8%) |
| Supervision | 0 | 3 (3.8%) | 2 (14.3%) | 1 (3.4%) |
| Regulatory activities | 10 (18.2%) | 23 (28.8%) | 2 (14.3%) | 9 (31.0%) |
| Other human-human interfaces | 2 (3.6%) | 8 (10.0%) | 0 | 2 (6.9%) |

**Table A3.** *Cont.*

| Parent Node/Child Node | Open—All Acceptable (55) | Closed—All Acceptable (80) | Open—All Unacceptable (14) | Closed—All Unacceptable (29) |
|---|---|---|---|---|
| **Liveware** | - | - | - | - |
| Experience, knowledge, and recency | 5 (9.1%) | 16 (20.0%) | 6 (42.9%) | 10 (34.5%) |
| Human physiology | 2 (3.6%) | 7 (8.8%) | 0 | 0 |
| Personal workload management | 7 (12.7%) | 10 (12.5%) | 1 (7.1%) | 4 (13.8%) |
| Personal physical or sensory limitations | 5 (9.1%) | 3 (3.8%) | 0 | 2 (6.9%) |
| Psychological limitations | 5 (9.1%) | 3 (3.8%) | 1 (7.1%) | 2 (6.9%) |

**Table A4.** Risk mitigation themes: NTSB reiterations. Recommendations consistently included two or more coded risk mitigation themes; therefore, percentages exceed 100%. Percentages (%) are specific to each column.

| Parent Node/Child Node | No Reiterations (172) | One Reiteration (10) | Two or More Reiterations (5) |
|---|---|---|---|
| **Software-Liveware (S-L)** | - | - | - |
| Human interface-procedures | 48 (27.9%) | 1 (10.0%) | 0 |
| Human interface-training | 38 (22.1%) | 1 (10.0%) | 0 |
| Other human-system support issues | 52 (30.2%) | 2 (20.0%) | 4 (80.0%) |
| **Hardware-Liveware (H-L)** | - | - | - |
| Human-hardware interface | 34 (19.8%) | 2 (20.0%) | 0 |
| Inadequate information/data sources | 26 (15.1%) | 4 (40.0%) | 3 (60.0%) |
| Human-software/firmware interface | 17 (9.9%) | 3 (30.0%) | 1 (20.0%) |
| Automation/automatic systems | 7 (4.1%) | 1 (10.0%) | 0 |
| Automatic defenses/warnings | 15 (8.7% | 1 (10.0%) | 0 |
| Operational material | 25 (14.5%) | 0 | 1 (20.0%) |
| **Environment-Liveware (E-L)** | - | - | - |
| Physical environment | 58 (33.7%) | 3 (30.0%) | 1 (20.0%) |
| Psychosocial factors | 1 (0.6%) | 0 | 1 (20.0%) |
| Company, management, manning or regulatory... | 96 (55.8%) | 6 (60.0%) | 4 (80.0%) |
| Operational task demands | 31 (18.0%) | 2 (20.0%) | 0 |
| **Liveware-Liveware (L-L)** | - | - | - |
| Communications | 16 (9.3%) | 1 (10.0%) | 0 |
| Team skill/CRM | 14 (8.1%) | 1 (10.0%) | 0 |
| Supervision | 5 (2.9%) | 0 | 1 (20.0%) |
| Regulatory activities | 38 (22.1%) | 3 (30.0%) | 4 (80.0%) |
| Other human-human interfaces | 11 (6.4%) | 0 | 1 (20.0%) |
| **Liveware** | - | - | - |
| Experience, knowledge, and recency | 38 (22.1%) | 2 (20.0%) | 0 |
| Human physiology | 9 (5.2%) | 0 | 0 |
| Personal workload management | 21 (12.2%) | 2 (20.0%) | 0 |
| Personal physical or sensory limitations | 10 (5.8%) | 0 | 0 |
| Psychological limitations | 11 (6.4%) | 0 | 0 |

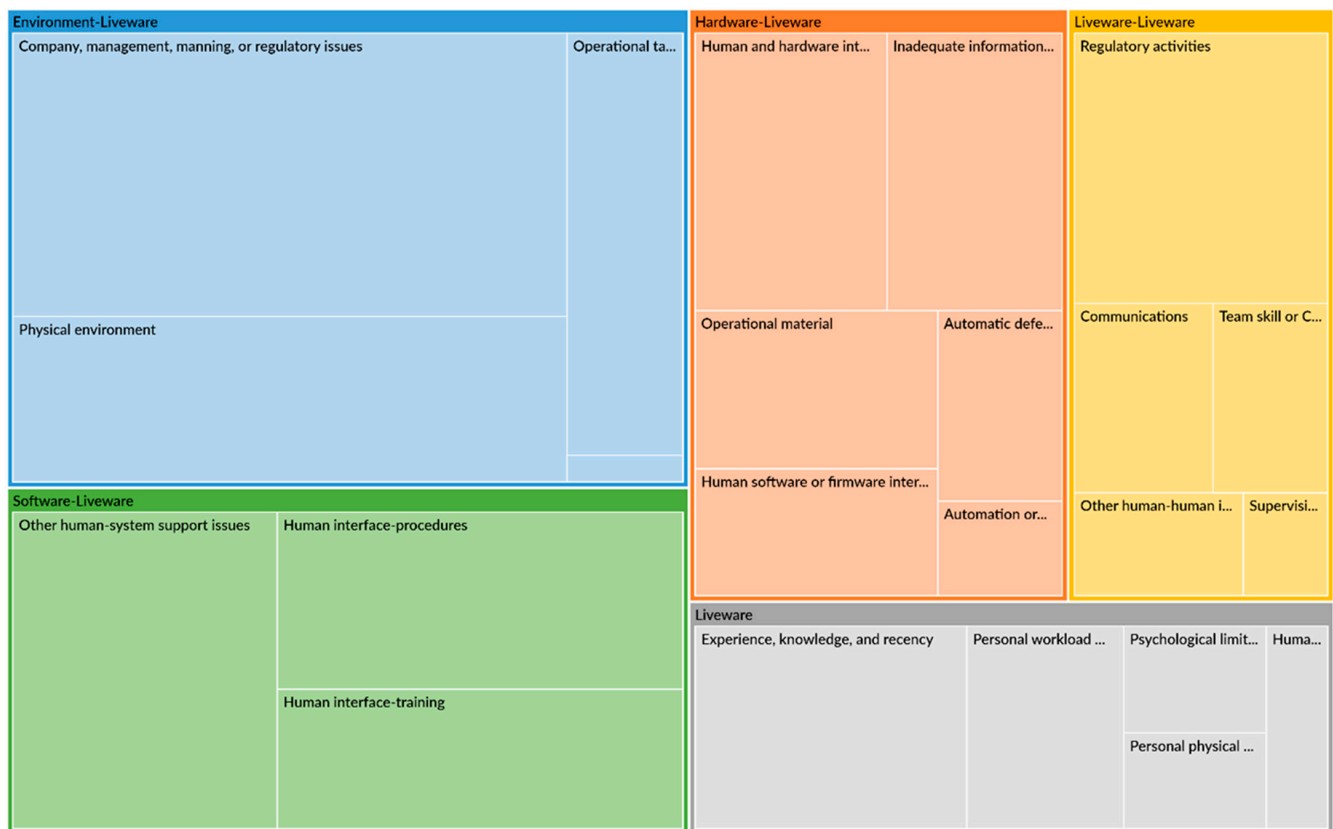

**Figure A1.** NVivo node hierarchy chart. This figure provides a relational visualization for the data contained in Table A2. Each parent and child node grouping is color coded.

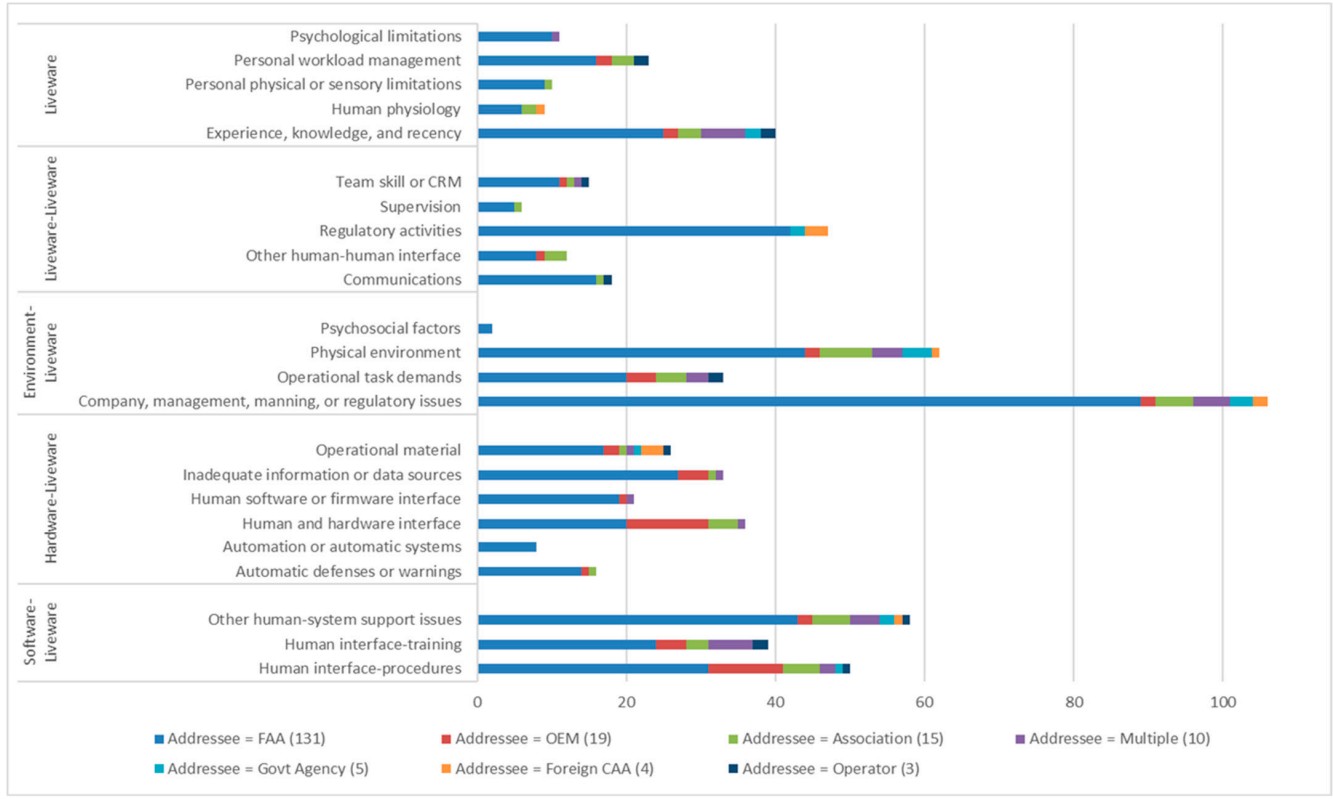

**Figure A2.** Risk mitigation themes: addressee distribution. The legend includes the number (value) of recommendations issued to each addressee.

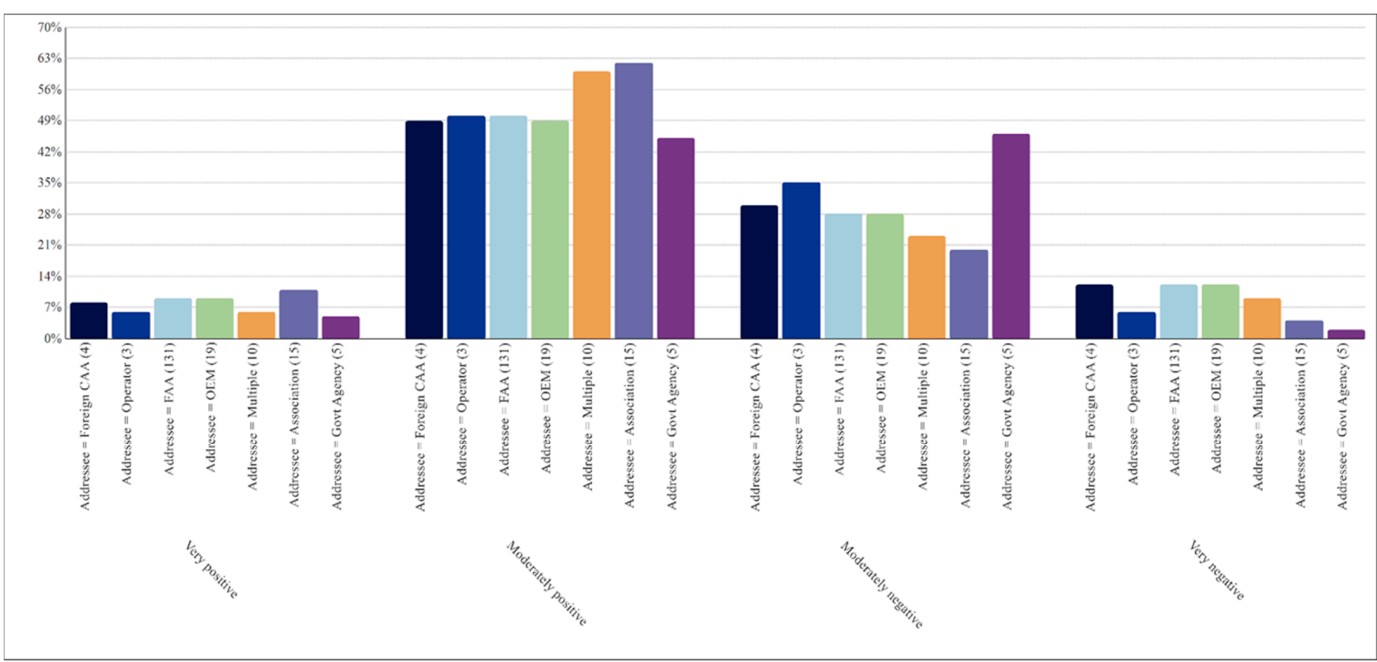

**Figure A3.** Sentiment analyses: addressee distribution.

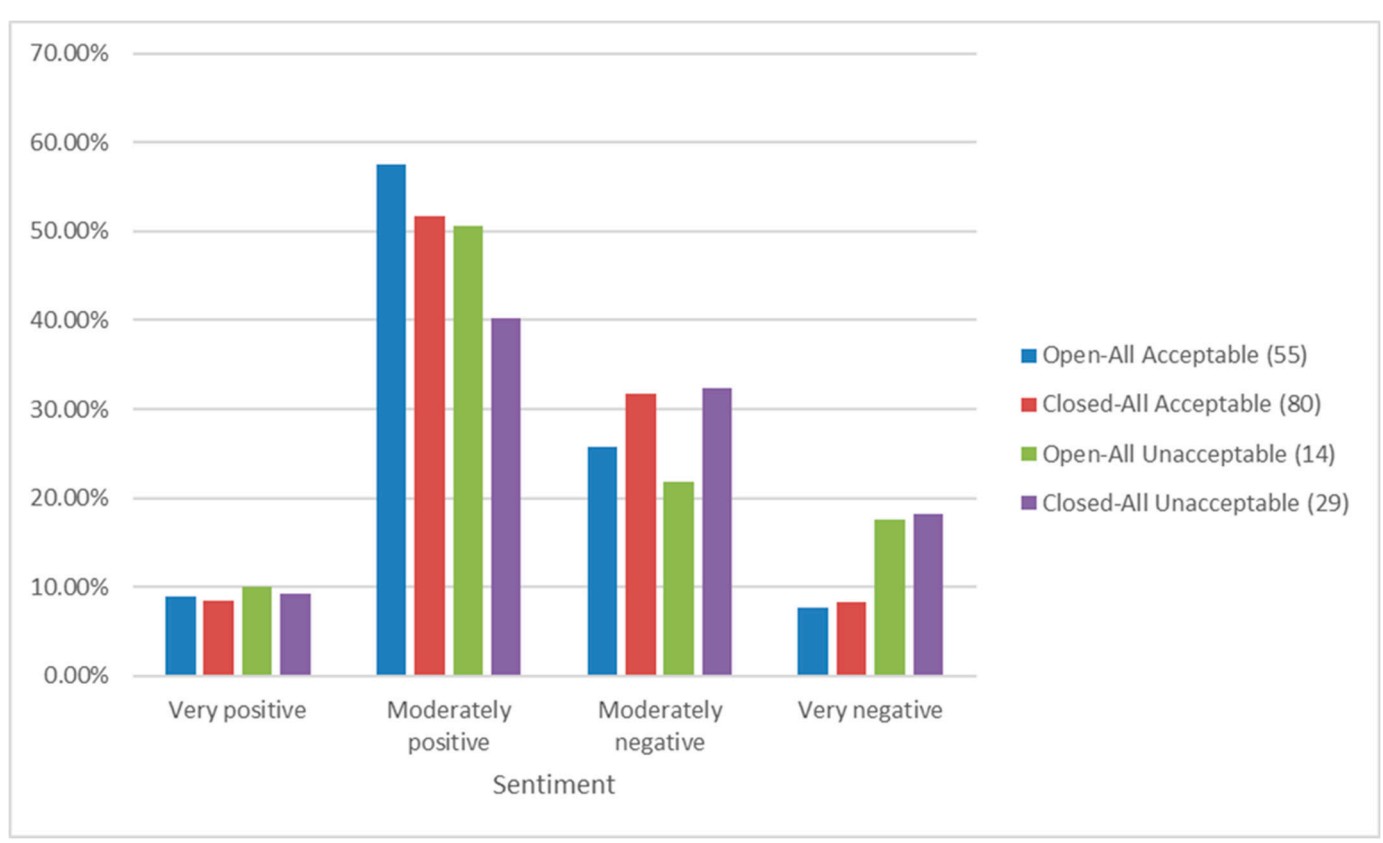

**Figure A4.** Sentiment analyses: status. Excludes reconsidered and no longer applicable statuses. Acceptable statuses include acceptable alternate and exceeded actions. Unacceptable statuses include superseded.

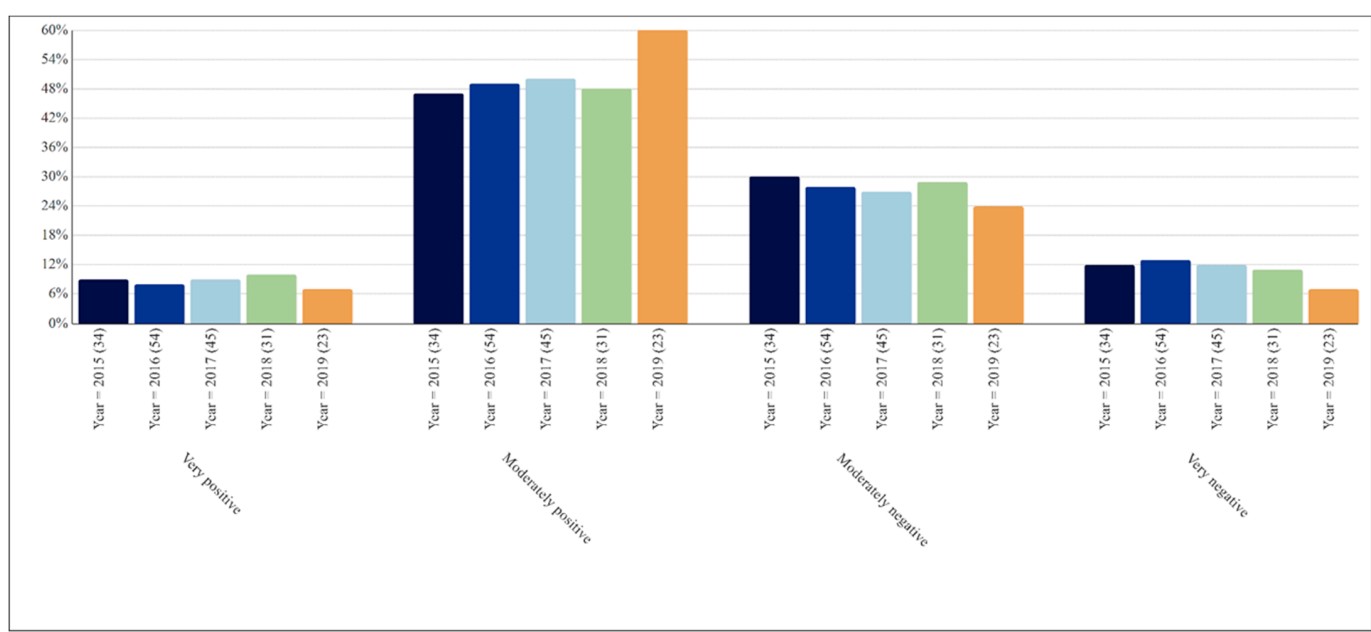

**Figure A5.** Sentiment analyses: annual distribution. The calendar year represents the year of the recommendation's issuance and each of its ensuing correspondences, regardless of the subsequent correspondences' associated dates.

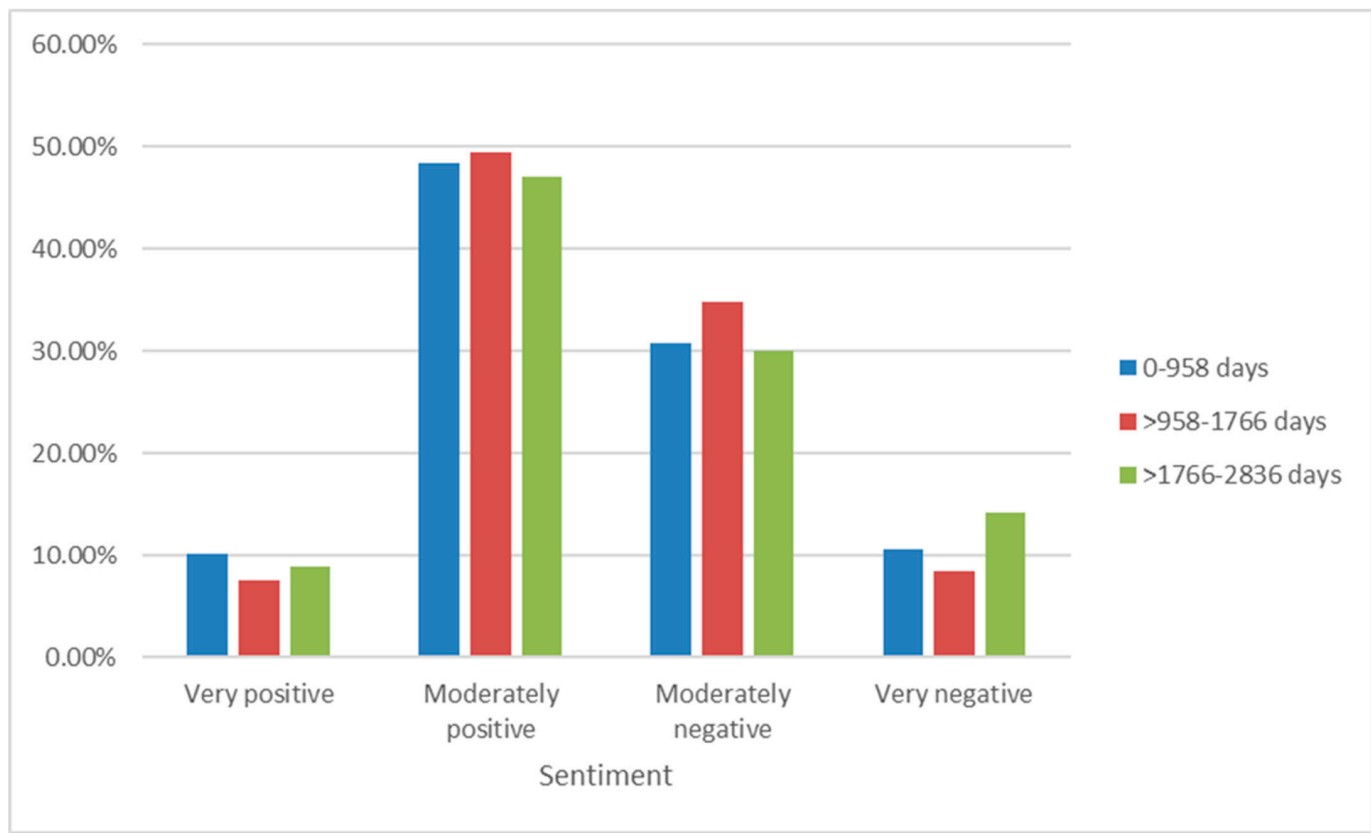

**Figure A6.** Sentiment analyses: days between recommendation issuance and closure. Excludes all open recommendation statuses. Short duration (0–958 days) included 39 closed recommendations; medium duration (>958–1766 days) included 40 closed recommendations; long duration (>1766–2836 days) included 39 closed recommendations.

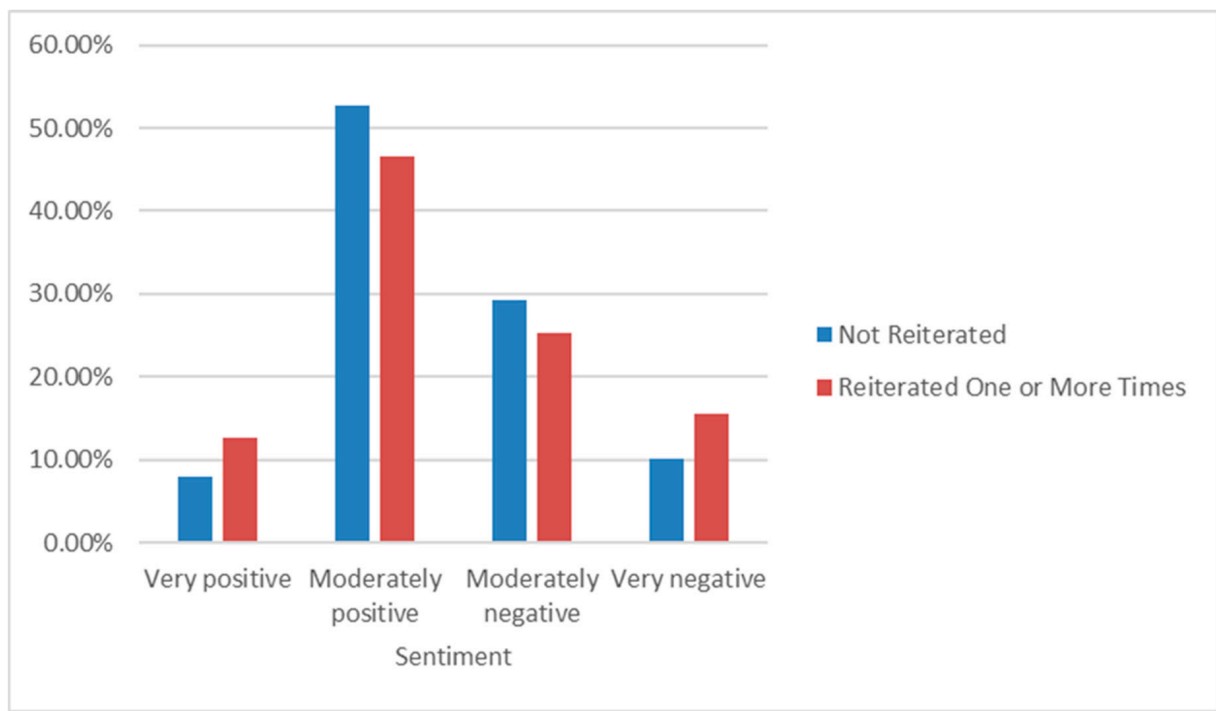

**Figure A7.** Sentiment analyses: reiterated recommendations. Any reiterated NTSB recommendations are grouped into "reiterated one or more times".

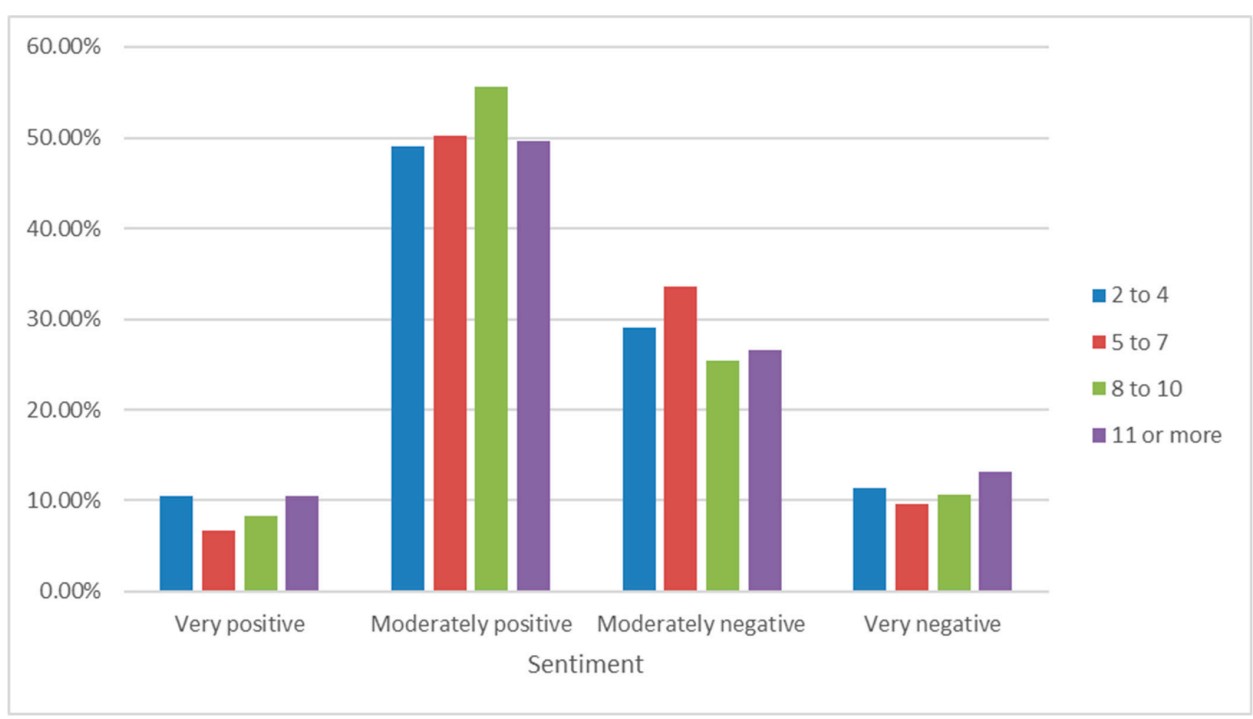

**Figure A8.** Sentiment analyses: correspondence count distribution.

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
