# Peer review of "Exploring National Transportation Safety Board Aviation Modality Recommendations Through Content and Sentiment Analyses: 2015–2019"

_safety, 2015_

Round 1

Reviewer 1 Report

Comments and Suggestions for Authors

Attached.

Reviewer 2 Report

Comments and Suggestions for Authors

Overall, this is an interesting paper with meaningful results. The paper presents a qualitative analysis, which is appropriate for the topic.  I personally believe that there is no way to get to the same results and understanding with a quantitative analysis given the data available, but there will be some who will want the more convincing evidence that a quantitative analysis would provide.

Given that this -is- a qualitative analysis, an consensus framework for analyzing the documents is imperative. The paper looks primarily at sentiment analysis. The authors use NVivo software. There is no discussion about how the NVivo software does its analysis. The authors describe that they manually go through to validate and modify NVivo's classifications. 

There are several natural limitations to this sort of qualitative study, and the authors honestly point out many of them: e.g. single researcher for coding, dependency on descriptive statistics and validations.

I find the use of the NVivo tool acceptable, but sentiment analysis by its nature is subjective. A description of the means by which NVivo automates the process so that it is clear that the authors understand how it happens is, in my opinion, required as part of the methodology section. Furthermore, a detailed description of the validation of NVivo's results, an explanation of the mental criteria the coder used to change NVivo's coding, and statistics on the number of such changes (both absolute and relative numbers) is also required for the methodology section.

The word cloud adds nothing to the paper and should be removed.

Reviewer 3 Report

Comments and Suggestions for Authors

This study contributes to the academic literature by addressing the underexplored areas of recommendation implementation processes and stakeholder interactions through content and sentiment analyses of NTSB aviation safety recommendations. The research design demonstrates methodological rigor, with reliable data sources and an innovative application of the SHELL model as the analytical framework. While the study offers valuable insights, enhancements could be made in the depth of the literature review, granularity of methodological descriptions, contextual interpretation of results, and adherence to disciplinary formatting standards.

  1. The abstract should explicitly highlight the study’s novelty, such as "the first systematic analysis of emotional interaction patterns in NTSB safety recommendations." Keywords should include "government communication" or "stakeholder engagement" to better encapsulate the research scope.
  2. In the Keywords, please Add "government communication" or "stakeholder engagement" to more comprehensively reflect the research theme.
  3. Supplement by explaining how the implementation efficiency of NTSB recommendations impacts aviation safety policies.
  4. Current literature lacks analysis of the emotional dynamics in recommendation communication; this study addresses this by examining sentiment as a critical factor in stakeholder engagement.
  5. Expand discussions on sentiment analysis applications in government agencies, citing parallel studies in healthcare or environmental policy domains.
  6. Explain the rationale for adopting the SHELL model over alternatives (e.g., HFACS), with examples of how ICAO ADREP coding criteria were adapted during data classification.
  7. Specify that inter-rater reliability was assessed through double-coding of 20% randomly selected samples, using Cohen’s κ to quantify agreement and resolve discrepancies via expert consensus.
  8. Acknowledge that NVivo’s automated coding may misinterpret contextual nuances (e.g., irony), mitigated by manual review of 15% of coded data to refine emotional categorizations.
  9. Incorporate key insights from Appendix Figures A1–A8 (e.g., risk theme distributions, emotional polarity differences) into the main text with inline figures for readability.
  10. The higher rejection rate for "corporate/regulatory" recommendations may correlate with institutional conflicts of interest or high compliance costs, warranting further qualitative exploration.
  11. Findings suggest that NTSB could optimize recommendation phrasing (e.g., emphasizing shared safety goals) to enhance acceptance by stakeholders like airlines and regulatory bodies.

12.The 2015–2019 dataset may not capture post-2020 aviation safety shifts (e.g., pandemic-related operational changes).

  1. The analysis of 187 recommendations limits generalizability to broader NTSB communication patterns, calling for replications with larger datasets.
  2. Ensure uniform journal name abbreviations (e.g., Risk Analysis in full or standardized abbreviation) and complete DOI URLs (e.g., https://doi.org/xxx).
  3. Simplify complex sentences (e.g., "Among 187 recommendations, the FAA received 70.1%, but 29 were rejected due to regulatory constraints") and use "recommendation recipients" consistently instead of alternating terms like "addressees."

Reviewer 4 Report

Comments and Suggestions for Authors

this manuscript makes a valuable contribution to the field of aviation safety and safety communication. It fills a noticeable gap in the literature and offers a useful foundation for more in-depth investigations into how safety recommendations are communicated and acted upon

However, the following can be improved:

Add a limitation section at the end

Include more recommendations

You need to make the title for figures 1, 2, 3 etc much clear, do not make it general for all tables, each one must be specific and when you read it, you understand what it is about.

Reviewer 5 Report

Comments and Suggestions for Authors

The paper illustrates application of a modern software to contexts analysis. The methodology is correct and according to good research practices. Focusing on aviation safety may attract readers not only from this field. The results are comprehensive and convincing.

However it is not clear what is the use of the results presented ? Who is “a customer” willing to make profit from or “implement” the results of the study ? What are Author's  recommendations from analysis of NTSB recommendations  ?

Round 2

Reviewer 2 Report

Comments and Suggestions for Authors

This version addresses my previous concerns and comments.

Reviewer 3 Report

Comments and Suggestions for Authors

All comments have been answered

Reviewer 5 Report

Comments and Suggestions for Authors

None.